# Correlative fluorescence microscopy, transmission electron microscopy and secondary ion mass spectrometry (CLEM-SIMS) for cellular imaging

Felix Lange[1,2☯], Paola Agüi-Gonzalez[3,4☯], Dietmar Riedel[5], Nhu T. N. Phan[3,4], Stefan Jakobs[1,2,4]*, Silvio O. Rizzoli[3,4]*

1 Research Group Mitochondrial Structure and Dynamics, Max Planck Institute for Biophysical Chemistry, Göttingen, Germany, 2 Clinic for Neurology, University Medical Center Göttingen, Göttingen, Germany, 3 Department of Neuro- and Sensory Physiology, University Medical Center Göttingen, Göttingen, Germany, 4 Center for Biostructural Imaging of Neurodegeneration, University Medical Center Göttingen, Göttingen, Germany, 5 Laboratory of Electron Microscopy, Max Planck Institute for Biophysical Chemistry, Göttingen, Germany

☯ These authors contributed equally to this work.
* sjakobs@gwdg.de (SJ); srizzol@gwdg.de (SOR)

**Data Availability Statement:** The data underlying the results presented in the study are available

## Abstract

Electron microscopy (EM) has been employed for decades to analyze cell structure. To also analyze the positions and functions of specific proteins, one typically relies on immuno-EM or on a correlation with fluorescence microscopy, in the form of correlated light and electron microscopy (CLEM). Nevertheless, neither of these procedures is able to also address the isotopic composition of cells. To solve this, a correlation with secondary ion mass spectrometry (SIMS) would be necessary. SIMS has been correlated in the past to EM or to fluorescence microscopy in biological samples, but not to CLEM. We achieved this here, using a protocol based on transmission EM, conventional epifluorescence microscopy and nano-SIMS. The protocol is easily applied, and enables the use of all three technologies at high performance parameters. We suggest that CLEM-SIMS will provide substantial information that is currently beyond the scope of conventional correlative approaches.

## Introduction

Cellular structure and function are currently investigated by a variety of imaging techniques, with resolutions ranging from sub-nanometer to millimeters. The best approaches to understanding cellular structure are typically connected to the use of electron microscopy (EM), in which electron-dense cellular elements are visualized with high precision. A downside to this approach is that specific organelles are identified only based on their morphology in the wide majority of the studies, since most EM applications are performed without labeling the organelles in a specific fashion. Specific labeling approaches are possible, in the form of immuno-EM, but are technically difficult, and require extensive optimization [1–3]. Moreover,

from Figshare (DOI: 10.6084/m9.figshare.
14416547.v1).

**Funding:** The work was supported by the German Research Foundation (Deutsche Forschungsgemeinschaft, DFG) through grants SFB1286/A05 to S.J., SFB1286/A03 to S.O.R. and SFB1286/B01 to N.T.N.P. The work was also supported by the Swedish Research Council through VR grant 2016-06800 to N.T.N.P.

**Competing interests:** The authors have declared that no competing interests exist.

substantial problems can be encountered by the use of antibodies for this type of labeling, which can only be avoided by very careful testing. This issue has become increasingly evident over the last years, albeit additional solutions may come from specially designed antibodies [4, 5].

In contrast, optical microscopy, and especially fluorescence microscopy, rely on specific labeling, based either on genetically encoded fluorophores or on affinity probes. This results in a precise localization of the proteins or organelles of interest, which enables an optimal analysis of their function, albeit in the absence of information on the cellular structure. Combining the two approaches, in the form of correlated light and electron microscopy (CLEM), unites the advantages of both technologies [6], enabling the analysis of specific elements in the context of the EM-described cell structure. This procedure is especially powerful when super-resolution fluorescence microscopy is employed [7], approaching the precision of well-optimized immuno-EM approaches.

Nevertheless, CLEM is still unable to cover an important aspect of cell biology, regarding cellular composition and its changes. Although cells maintain the function and morphology of their organelles largely constant, they continually replace older components (*e.g.* proteins) with newly synthesized ones, to avoid the accumulation of damaged molecules. This process takes up from several hours to a few days in cell cultures [8], and days to weeks *in vivo* [9]. This process is difficult to analyze by electron or light microscopy, and is therefore typically studied by biochemical tools, as mass spectrometry. The samples are pulsed with molecular precursors (*e.g.* amino acids) that carry rare stable isotopes, with the most popular being $^{15}$N and $^{13}$C. The isotopes are incorporated in the newly synthesized molecules, which are then identified by mass detectors, thereby enabling the investigation of protein turnover. Such experiments have been performed in many types of cell cultures [8, 10–12], or even *in vivo* [9, 11, 13].

These approaches, however, lack spatial precision, since the cellular samples need to be homogenized and biochemically processed. The solution to this issue has been the implementation of nanoscale secondary ion mass spectrometry (nanoSIMS), in which the isotopes are localized at a resolution comparable to that of light microscopy. Most nanoSIMS experiments in biology rely on the use of a $Cs^+$ beam that is scanned across the surface of the sample, and causes the sputtering of secondary particles. A large proportion of the particles are ionized, and are guided by ion optics to several mass detectors, which provide quantitative assessments of their abundances, in an image format. The lateral resolution of the images lays at around ~50–100 nm in the X-Y plane [11, 13]. The axial (depth) resolution depends mainly on the depth from which the secondary ions are obtained, and can be as low as 5–10 nm in biological specimens [14, 15]. NanoSIMS has been employed in several experiments testing cellular turnover, for example by feeding cells or animals with metabolites containing $^{15}$N isotopes [16–18].

The identification of specific cellular structures in nanoSIMS has been difficult, since organelle morphology is difficult to detect directly (albeit large organelles as the nucleus can be identified from most conventional nanoSIMS images). Several efforts have been made to enable the labeling of specific proteins in SIMS, either by using genetically encoded probes that are revealed by compounds carrying rare isotopes [14, 19], or by affinity labeling with isotopically-modified probes [20, 21]. None of the probes, however, are currently commercially available, which implies that the most popular way to identify structures has been to combine SIMS with EM [22–24] or with fluorescence microscopy, either using conventional or super-resolution implementations [14, 15, 18].

These correlation approaches reflect the respective shortcomings of the techniques employed, being limited to revealing structures (EM) or specific proteins (fluorescence

microscopy). A simple solution would be to combine nanoSIMS with both fluorescence and EM, in the form of correlative CLEM-SIMS, to enable the visualization of the cellular structure (EM), of specific proteins (fluorescence microscopy) and of the cell composition (nanoSIMS). This approach has not been implemented, due to unresolved difficulties in implementing all of the three techniques on a single biological sample. We have generated a suitable CLEM-SIMS protocol here, based on transmission electron microscopy (TEM), conventional epifluorescence imaging, and nanoSIMS, and we applied it to cell culture preparations (**Fig 1**). This protocol is, after initial optimization, straightforward in its application, and should enable optimal analyses of cell composition and turnover in the future.

## Results

### CLEM implementation

To establish a robust CLEM-SIMS protocol, we relied on existing high-accuracy CLEM approaches [25]. To this end, and to ensure superior ultrastructural preservation of cellular and organelle membranes, we used high-pressure freezing (**Fig 1**). This approach has the additional benefit of maintaining the fluorescence of many fluorescent proteins and organic fluorophores [26, 27].

Concretely, we transfected HeLa CCL cells with mito-mCitrine plasmid to label the mitochondria. Subsequently, cells were immobilized by means of high pressure freezing followed by freeze substitution and UV light-aided resin curing in the cold. 160 nm-thick sections were

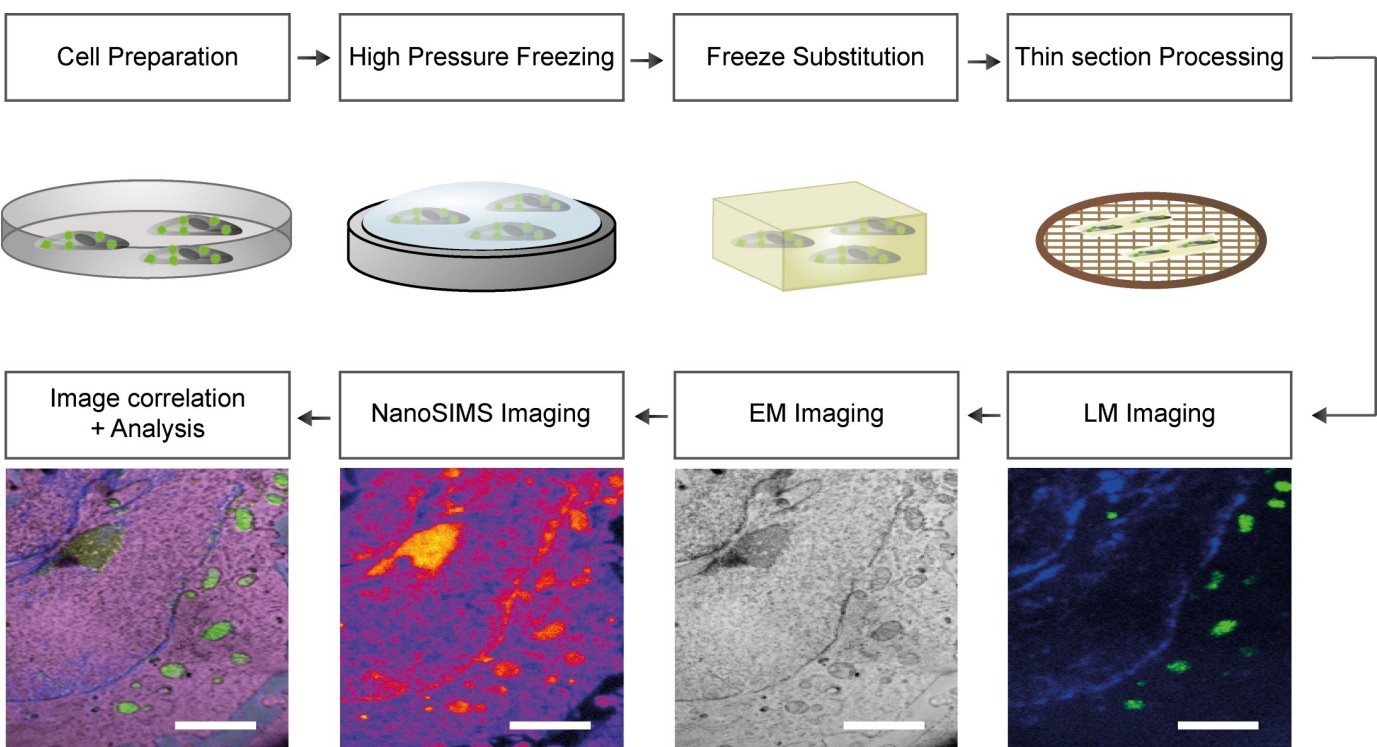

**Fig 1. General overview of the workflow for CLEM-SIMS imaging.** The cells are cultured following standard protocols, and fluorescence labelling can be applied to address specific questions. The cells are then immobilized by high pressure freezing, followed by freeze substitution. After the polymerization of the resin, the blocks are sectioned and the slides are placed on electrically conductive and referenced grids. Once the thin-sections are placed on grids, the first imaging step is fluorescent microscopy, followed by TEM, and finally by nanoSIMS. When the same areas have been scanned/imaged by the three techniques, the images are processed for registration and are analyzed. Scale bars: 3 μm.

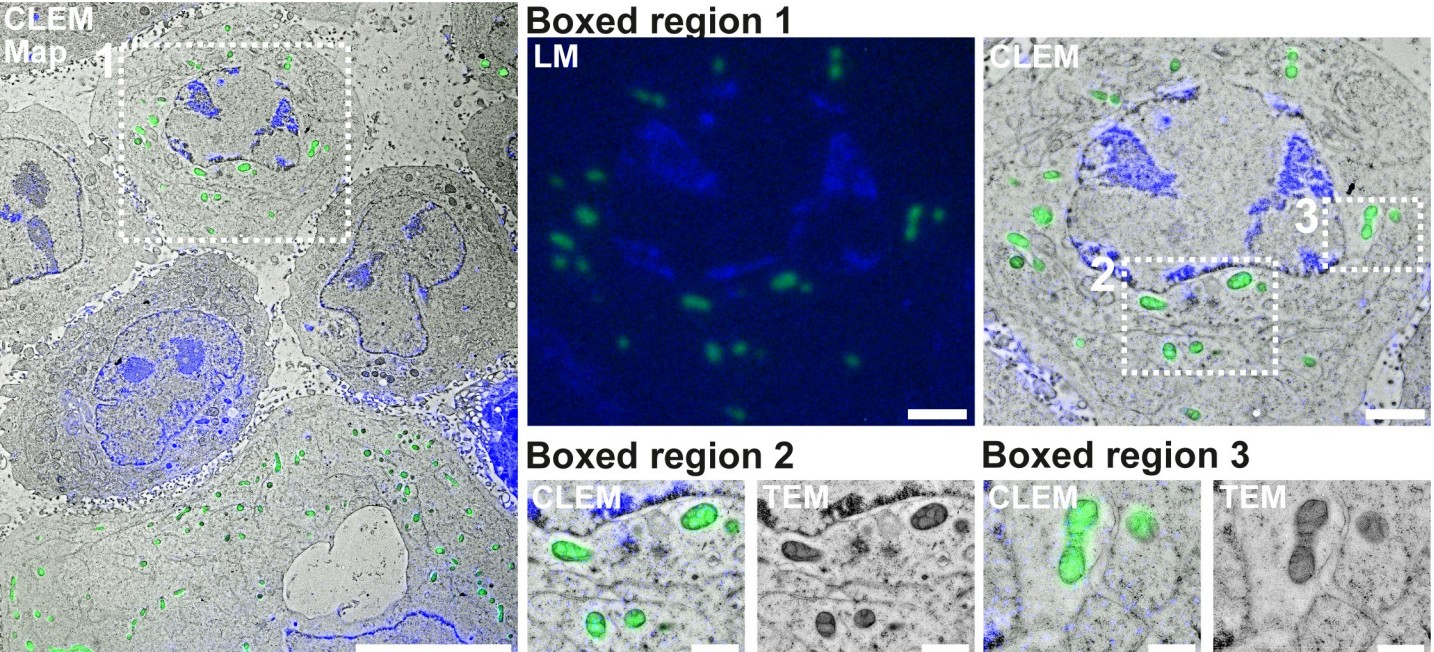

**Fig 2. A large-view CLEM image.** A large field was imaged in both TEM and fluorescence, relying on the DAPI (blue) channel, which shows the uranyl acetate fluorescence, and on the mito-mCitrine channel (shown here in green). Scale bar: 10 μm. Boxed region 1 shows a higher zoom view on the light microscopy image as well as the correlated image with TEM of a cell marked by a white square in the overview image. Scale Bar: 2 μm. Boxed regions 2 and 3 show detailed images of CLEM and TEM in the regions of interest. Scale bar: 1 μm.

then prepared at room temperature from the resin blocks and mounted on TEM finder grids. The use of finder grids was essential, as it enabled us to correlate readily the positions of the cells imaged in the different technologies. For an initial CLEM-map of the samples, we placed the TEM grids on glass slides, covered them with a drop of buffered solution (PBS at pH 7.4), overlaid them with a glass coverslip, and analyzed them by fluorescence microscopy. Regions of interest were identified using a low magnification objective, and subsequently higher resolution images were recorded using an oil immersion objective (100x).

We recorded the mito-mCitrine fluorescence (shown in green), together with in the DAPI channel (shown in blue), which essentially is the fluorescence signal from the uranyl acetate, which aids to correlate the fluorescence and EM images. In addition, the TEM-finder grid boxes were imaged in the brightfield mode.

After recording of the light microscopy images, the grids were manually recovered from the glass slides, washed, dried, and subsequently imaged by 2D-TEM. A final overlay of the TEM and fluorescence images was performed in ICY, using the plug-in eC-CLEM [28] (**Fig 2**).

For the following CLEM-SIMS experiments, we incubated HeLa CCL cells with Mito-Tracker Deep Red FM to label the mitochondria. The preparation of the cells as well as the analysis via light and electron microscopy followed the exact procedure as described above. Registered images were used as a map to guide subsequent NanoSIMS measurements.

## CLEM-SIMS

The combination of EM and fluorescence imaging brings substantial information on cellular structure and function. However, it provides very limited information on cellular composition. The cellular chemical composition is of substantial interest not only in relation to the positioning of different cellular elements, but also in relation to cellular turnover, as discussed in the

Introduction. We note, that the turnover of some cellular elements, such as proteins, can also be analyzed by fluorescence methods, in which the cells are pulsed with non-canonical amino acids that are incorporated in newly synthesized proteins, and are later revealed by specific chemical reactions [29]. This procedure, however, only reveals the newly-synthesized proteins, while SIMS approaches can provide the balance between general protein abundance and new synthesis, resulting in higher precision in the interpretation of turnover [30].

We therefore proceeded to image the samples in SIMS, relying on the nanoSIMS implementation, as described recently [19, 20]. The samples were mounted in a nanoSIMS instrument, and were first inspected using an in-built optical camera (CCD), to find the area of the grid that had been imaged in TEM, relying on the location markers inscribed on the finder grids. We then switched to the SIMS mode, and used a secondary electron detector to confirm the exact imaging position. The electron detector indicates the positions of the grid bars with precision, and also shows the positions of the cells, as their thickness in the section is slightly higher than that of the surrounding plastic resin.

A short implantation with relatively low current (15 pA) was then applied, to enhance the signal coming from the sample and to reach a stable secondary ion yield from the sample (a status known as steady state). The thin samples could not be treated with high currents, as they otherwise would have been damaged, especially when aiming to image smaller areas than ~20x20 μm, or when implanting for long periods. Using a current of 15pA for ~1 minute was sufficient to visualize the cells and to reach the steady state of the surface.

To optimize spatial resolution, the primary ion beam $Cs^+$ was adjusted using the smallest possible aperture, thereby reducing the beam diameter, and increasing resolution. However, since this will also reduce the primary ion current leading to the reduction of the secondary ion signal intensity, some optimization should be performed, especially for biological samples, which contain multiple isotope species with different ranges of abundance and ionization yields, and where a balance needs to be found between the spatial resolution for imaging and the sensitivity required to detect low signal species. It is important to note that the probability of ionization of a molecule is typically lower than $\sim 10^{-4}$, which implies that, even if the detection limit of the nanoSIMS device is in the range of parts per million, the available amount of analytes might be too low to obtain a good signal at high resolution [31]. To image the cells of the first experiment (transfected with mito-mCitrine), we applied a primary ion current of ~1.15pA, selecting D1:3 aperture. Since some samples collapsed during the imaging procedure, for the following experiment (cells incubated with MitoTracker Deep Red FM) we employed a smaller aperture (D1:4) and a current of 0.5pA. With these settings we achieved a satisfactory spatial resolution, while also obtaining a good signal-to-noise ratio from all of the targeted ions.

In experiments that include isotope enrichment or aim to study elements with higher natural abundance, the aperture slits and the current of the primary beam could be further reduced. Moreover, one can improve the visualization of low signal species by collecting multiple images on the same area. For each image, a thin layer (of a few nanometers) is ablated by the procedure [15, 32]. One can then sum the resulting images. This reduces the axial (depth) resolution, since multiple images of thin sample layers are combined into a single image, as thick as the sum of all imaged layers. This disadvantage is more than compensated by the fact that the summed image contains higher levels of all measured isotope species, and has therefore a much improved signal-to-noise ratio. The maximum depth that can be analyzed will be equivalent to the section thickness. We optimized this parameter to 100–160 nm in our experiment, which is convenient for both TEM and nanoSIMS. Thinner specimens provided improved TEM resolution, but could not be imaged in nanoSIMS, as they were readily damaged by the $Cs^+$ beam. It was also found that the grid treatment could be fine-tuned to improve the stability

of the samples, with copper grids with a support film of 10 nm Formvar and 1 nm Carbon (obtained from a commercial provider; see Methods) being the most stable.

As the aim of this work was to test the feasibility of the CLEM-SIMS correlation, we did not introduce exogenous elements to detect the turnover of individual structures in SIMS. To nevertheless demonstrate the possibility of such experiments, we analyzed the TEM grids for multiple isotopes, including the rare stable isotopes that would be used in turnover experiments, such as $^{13}C$ or $^{15}N$. Carbon isotopes were measured in the form of $C_2^-$ ions, while nitrogen isotopes were measured as $CN^-$ ions. As expected, the dominant carbon isotope, $^{12}C$, provided a clear overview of the cells (Fig 3), while the rare isotope $^{13}C$ was detected far more poorly, according to its natural abundance (~1.1%). A similar view was obtained for nitrogen isotopes (Fig 3), while relatively clear images were obtained for $^{31}P$ and $^{32}S$, which are reasonably abundant in biological samples.

To analyze the precision of our image registration, we drew lines on mitochondria that were visible in all three imaging modalities, and measured the full-width-at-half-maximum (FWHM) of the respective signals. The resulting values were similar for the three imaging modalities (S1 Fig).

The resolutions that could be obtained in the three technologies is typical for the technologies involved (S1 Fig). For SIMS, a widespread manner to calculate the lateral resolution is to measure the distance across which the signal drops from 84% to 16% of its maximum [15]. An average resolution of ~160 nm was obtained, which is superior to the resolution measured in fluorescence microscopy, but is not as high as the resolution we obtained in similar biological samples placed on silicon wafers, around 80–110 nm [15]. This is explained by the need to adjust to the mechanical fragility of the TEM samples. The resolution obtained in fluorescence microscopy and EM was similar to the expected performance of these techniques in normal samples (S2 Fig).

## CLEM-SIMS analysis of cultured cells

We employed CLEM-SIMS to analyze the isotopic composition of several cellular elements that were visible in TEM, and of mitochondria visualized in fluorescence microscopy. Interestingly, not all mitochondria were labeled fluorescently, with several being evident in TEM, but displaying no MitoTracker labeling (arrowheads in Fig 4A). Presumably, the mitochondria lacking the fluorescence label were dysfunctional (although a visual inspection in EM did not indicate obvious morphology problems), and were therefore not marked by MitoTracker, which requires an intact mitochondrion, with a substantial membrane potential [33].

We analyzed mitochondria in several cells, as well as other organelles, including dense cellular granules, which are presumably similar to dense-core vesicles [34], and the euchromatin and heterochromatin areas from the nuclei. We also analyzed regions that did not apparently contain any specific organelles, and could therefore be regarded as cytosol-filled regions. We observed several significant differences between the different areas (Fig 4B–4D). For example, the heterochromatin contained higher levels of both $^{31}P$ and $^{32}S$ than the euchromatin (when normalized to the ubiquitous $^{12}C$). The same was observed when comparing the mitochondria and the granules to the cytosol. The granules and the heterochromatin were also especially rich in $^{14}N$ (Fig 4B), which suggests that these compartments are especially protein-rich, when compared to other cellular areas.

Finally, we relied on the combined information provided by EM and fluorescence imaging, to differentiate between functional (MitoTracker-labeled) and presumably non-functional mitochondria (not labeled by MitoTracker). As shown in Fig 4E and 4F, Mitochondria lacking MitoTracker exhibited substantially lower levels of $^{14}N$, when compared to the MitoTracker-

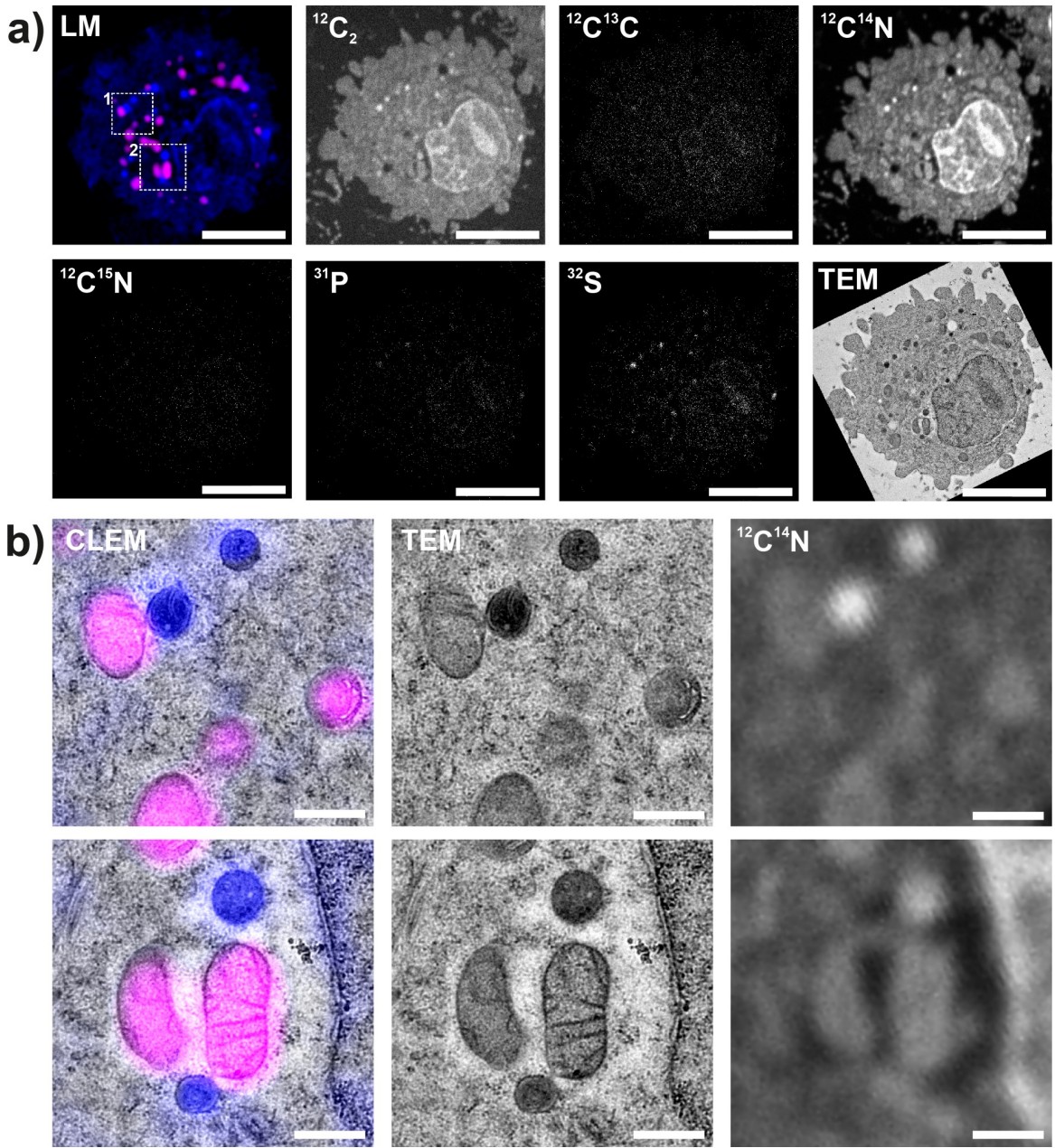

**Fig 3. Detailed views of CLEM-SIMS in cell culture.** a) An individual cell was imaged in light microscopy, TEM, and nanoSIMS. The fluorescence channels show the uranyl acetate fluorescence (blue) and the MitoTracker fluorescence (magenta). The measured isotopes are indicated in the different nanoSIMS images. Nitrogen isotopes ($^{14}$N, $^{15}$N) are measured as $CN^-$ ions, while carbon isotopes ($^{12}$C, $^{13}$C) are measured as $C_2^-$ ions. $^{13}$C and $^{15}$N are rare isotopes, which explains the low intensity of the respective images. Scale bars: 2 μm. b) Higher zoom views of the areas marked by white squares in panel A. Correlations of fluorescence (light) microscopy (left panels), TEM (middle panels) and nanoSIMS (right panels, $^{14}$N) are shown. Scale bars: 500 nm.

containing ones. It is unlikely that this is due to widely different levels of proteins or nucleic acids in these mitochondria, since both $^{31}$P and $^{32}$S levels were similar (**Fig 4F**). It is more likely that the mitochondria lacking MitoTracker have a perturbed metabolism, and therefore have lower levels of metabolites containing Nitrogen, but not Sulphur or Phosphorus, as nicotinamide, but this issue will require further testing. Overall, this experiment indicates that

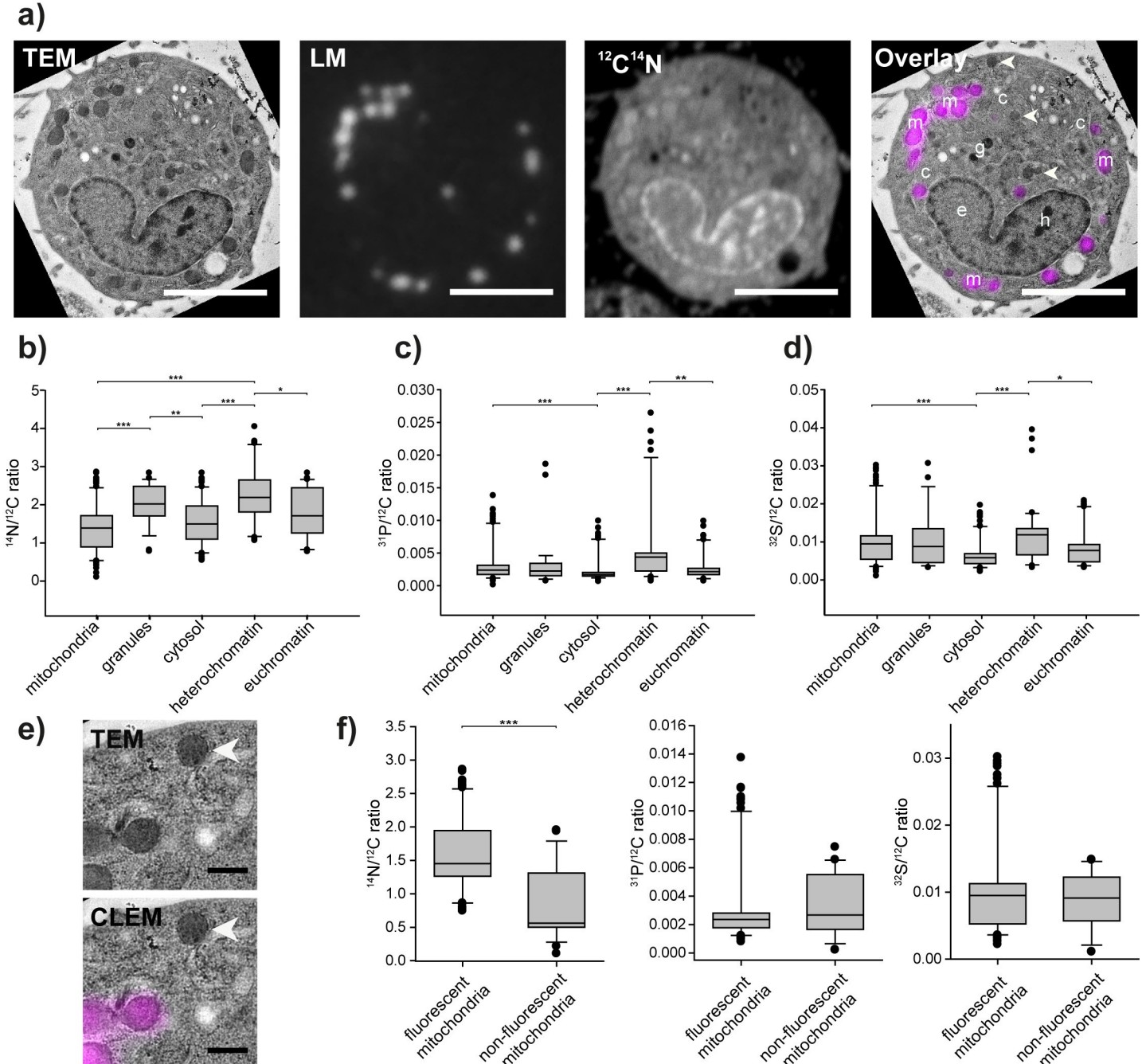

**Fig 4. An analysis of cultured cells with CLEM-SIMS shows different isotopic distribution in different cellular compartments.** a) CLEM-SIMS images of a cell labeled with MitoTracker (magenta). The overlay combines the TEM and fluorescence images. The arrowheads indicate several mitochondria not labeled by MitoTracker, but visible in TEM. Scale bars: 5 μm. The letters indicate mitochondria (m), granules (g), euchromatin (e), heterochromatin (h) and cytosol (c). b-d) NanoSIMS analysis of specific compartments in the cells, identified in the TEM images. The graphs show box plots from 34 to 89 cellular regions, from multiple analyzed cells. Ratios of different isotopes to the ubiquitous $^{12}$C (present in the cellular material and in the plastic resin) are shown. The middle line in the boxes indicates the median, while the boxes show the 25th percentiles; the error bars show the 75th percentiles, with outliers indicated by the symbols. Statistical differences were tested using Kruskal-Wallis tests, followed by Tukey post-hoc tests; * $p < 0.05$; ** $p < 0.01$; *** $p < 0.001$. b) $^{14}$N analysis. c) $^{31}$P analysis. d) $^{32}$S analysis. e) A higher zoom view of mitochondria labeled with MitoTracker (magenta) or lacking the label (white arrowhead). Scale bar: 500nm. f) Analysis of the fluorescently-labeled and non-labeled mitochondria. The measurements compare 85 labeled mitochondria and 24 non-labeled ones. Statistical differences were tested using Mann-Whitney tests; *** $p < 0.001$.

strong differences in the composition of these mitochondria can be observed in nanoSIMS, albeit they appear largely similar in EM.

## Discussion

The protocol discussed here enables the analysis of the sample structure, with excellent preservation of all organelles, while revealing the positions of specific proteins of interest, by fluorescence microscopy. The isotopic composition of the sample is then determined in SIMS, and can be correlated to the information from EM and fluorescence. This enables an overview of the basic composition of several cellular organelles, with a focus on abundant natural isotopes, as $^{14}$N, $^{12}$C, $^{31}$P or $^{32}$S. Several differences could be detected, some of which can be assigned to the known composition of the respective organelles, while others are more surprising, and may require further study.

It has generally been difficult to combine fluorescence microscopy with the optimal preservation of the samples for electron microscopy. We perform this here by high pressure freezing, followed by freeze substitution. Many other CLEM protocols rely on chemical fixation, followed by embedding in a plastic resin at room temperature. Such protocols are more flexible in the implementation of fluorescence labeling, since various immunolabeling tools can be employed, typically before embedding the samples. Post-embedding labeling has also been used [35], but is considerably less efficient. The main issue with using chemical fixation protocols is that they result in strong modifications to the samples, which are due to several artefacts, as explained in the following phrases. First, the sample is slowly dying, in a process that probably results in considerable biological changes, including aberrant biological activity or osmotic swelling [36]. Second, depending on the chemical fixation procedure, several components may not be fixed, and may move during the ensuing sample preparation steps [37, 38]. Third, pre-embedding immunolabeling typically requires sample permeabilization, which damages the cell morphology [39]. Many fixation protocols have been optimized in the last decades (see Richter et al., 2018 [40], and references therein), but none of these issues have been definitively solved in chemical fixation. One important procedure, however, has been to replace the antibodies used in the large majority of immunolabeling protocols, with nanobodies, which are substantially smaller, and can penetrate into fixed cells and tissues without a need for permeabilization [41]. This solves the major artifact-inducing step of the chemical fixation CLEM procedures, and therefore would allow them to be used in works such the one performed here (although the low number of available nanobodies still limits the wide application of this concept). Another solution for this issue would be the use of high-pressure freezing and freeze-substitution followed by rehydration, immunostaining using antibodies in the aqueous milieu, and finally embedding and processing for imaging. This approach has been used for immuno-EM in the past [42–44], and should be applicable also for CLEM procedures based on antibody immunostaining.

The nanoSIMS implementation of the CLEM-SIMS procedure is relatively straightforward. The sample thickness needs to be taken into consideration, as we could not obtain accurate SIMS images for samples of less than ~90 nm, irrespective of the resin employed, or of the nature of the underlying grid coating. As samples much thicker than ~160 nm will become a problem in conventional TEM imaging, the usable thickness interval is limited. These samples are relatively thick, from the TEM point of view, which implies that the grid coating should be as thin as possible, while ensuring good conductivity, which can be ensured by the addition of a thin carbon layer [24]. Difficulties in correlating the different images may be caused by sample drift during the SIMS imaging, which, unlike fluorescence or TEM imaging, takes many minutes per specimen. Minimizing drift is therefore an important aspect in this approach,

although it is probable that some level of image processing may be required for a perfect overlap. As several cell elements are observed well in both TEM and SIMS, including the cell borders or the nucleus, they can be used in overlaying the images, without the need for additional fiducial markers. Nevertheless, the addition of gold particles, which are easily observed in both TEM and SIMS, and are also visible in fluorescence microscopy [45], could be useful for increasing the overlay precision.

The use of several sources of information in CLEM-SIMS should enable higher precision in experiments investigating sample composition, and for cellular turnover. This has been the main application for SIMS in cell biology, both for cell investigations [15, 30] and for *in vivo* experiments [16, 17]. Typically, an isotopic amino acid was fed to the organisms, and the location of the newly-synthesized proteins, containing the isotopes, was then revealed by SIMS. This enabled the discovery of several cellular components with unusual turnover, or of previously unobserved links between organelle age and function [30]. An interesting perspective on turnover experiments has been provided by SIMS applications in which the animals were pulsed with isotopic food for a limited time period, followed by long chase times. Such experiments revealed "old" cells, with low turnover [22, 23]. Such cells were discovered in different organs, and have not yet been understood in detail. Alternatively, young, newly formed cells may be investigated [18], following similar experiments.

Such experiments would benefit from further information from CLEM-SIMS. For example, a correlation between fluorescence microscopy and SIMS suggested that synaptic vesicles, which are small neurotransmitter-filled neuronal organelles, are only able to function for a few days after their synthesis, with older vesicles no longer involved in releasing neurotransmitter [30]. However, the respective work could not provide any information on the morphology of the fluorescence-identified vesicles. It is possible that the older vesicles are confined in special compartments, and are thus prevented from functioning (see review by [46]). Alternatively, they may have fused to synaptic endosomes, resulting in changes to both their morphology and function [47]. Both hypotheses could be tested by CLEM-SIMS, although they could not be approached by fluorescence-SIMS alone. Applications using EM-SIMS correlations could also benefit from the current CLEM-SIMS protocol, by including the fluorescence perspective. For example, the recent discovery of both old and new cells in a variety of organs [22, 23] could be enhanced by experiments seeking to identify the key differences between the respective cells, which could be targeted by labeling specific markers fluorescently.

The results obtained here, albeit only aiming to showcase the potential of this technology, and not to address a specific biological question, nonetheless indicated several interesting features. Some observations were relatively simple, as the finding that heterochromatin contains higher levels of Phosphorus than euchromatin. This is explained by the higher levels of nucleic acids in heterochromatin, as the nucleic acids are substantially richer in Phosphorus than other cellular components. The similar finding that mitochondria are richer in Phosphorus than the cytosol could also be attributed to the presence of nucleic acids in these organelles. However, the similar observations that Sulphur is enriched in mitochondria and in the heterochromatin is less straightforward. Sulphur is present mainly in proteins, and only to low levels in other cellular elements, arguing that these compartments have particularly high densities of proteins. This could be investigated by specific labeling experiments in the future.

Other correlative approaches, such as correlation of electron microscopy and energy dispersive X-ray (EDX) or electron energy loss spectroscopy (EELS), have already been well established and are frequently applied in life sciences [48, 49]. These techniques use an electron beam to excite the electron in the atomic inner shell, to produce a characteristic X-ray or a loss of energy due to inelastic scattering of the electron. They enable the analysis of endogenous elemental composition of biological samples at nanoscale resolution, compatible to TEM.

These approaches add another dimension to identify the elemental structure of the cells and tissues, in addition to the morphological ultrastructure from the TEM. However, they exhibit several limitations [1, 2]. First, the compositional information obtained by EDX and EELS is limited to elements, and this does not enable a very high discrimination of the biological content from surrounding background, compared to nanoSIMS. The latter technique provides fragmented ions specifically for biological materials, such as CN-, CC-. Second, EDX and EELS can only detect a subset of elements, while NanoSIMS is able to detect almost all of them. Third, EDX and EELS are unable to analyze isotopically labelled samples, which can typically be measured by nanoSIMS. The latter therefore enables the study of cellular turnover via the analysis of the enrichment of exogeneous isotopic compounds, while EDX and EELS do not. These shortcomings of the correlative techniques that use EDX or EELS emphasize even more the need of a new correlative approach employing nanoSIMS.

At the same time, the type of analysis performed here also needs to be received with a note of caution, as in these experiments one can only analyze the materials that are left within the sample after processing. As the protocol involves the exchange of water to organic solvents and then to a plastic resin, substantial changes in ionic composition are expected, along with a substantial loss of lipids and small, non-fixable metabolites, although the general morphology will be preserved [44]. Another important issue is that the environment of the plastic resin may change the ionization of particular species, and therefore may affect the SIMS measurements in an unpredictable fashion. This effect should be minimized by embedding all experimental and control samples in the same resin, following identical protocols. Other potential improvements include fluorescence imaging at cryogenic temperatures, a procedure that increases dramatically the fluorescent signal coming from uranyl acetate [50, 51]. Furthermore, the use of glass coverslips in our study implies that the cells needed to be detached using trypsin. A more convenient procedure would be the use of sapphire disks, which removes the need for detaching the cells before the vitrification procedure.

Overall, in spite of these limitations, the technology introduced here should enable a better and more specific understanding of SIMS signals, by enabling their correlation to specific structures, observed in electron and fluorescence microscopy. This should enable a variety of new experiments, thereby bringing new insight in the field of cellular composition and turnover.

## Materials and methods

### Cell culture

HeLa CCL cells were cultivated in DMEM high glucose medium supplemented with gluta-MAX (Thermo Fischer Scientific, Waltham, MA, USA) further supplemented with 100 U/ml penicillin and 100 μg/ml streptomycin (Merk Millipore, Burlington, MA, USA), 1 mM sodium pyruvate (Sigma-Aldrich, Munich, Germany), and 10% (v/v) fetal bovine serum (Merck Millipore) in an incubator set to 37°C with 5% CO2.

### Cell transfections

For HeLa CCL transfection and mitochondrial labelling, we used mito-mCitrine plasmid with TurboFect (ThermoFisher Scientfic, Germany) according to the manufacturer's protocol. In detail, HeLa CCL cells were seeded on 10 cm cell culture dishes, one day before the transfection, at approximately 70% confluency. Prior to transfection, HeLa CCL were switched to DMEM medium without serum and antibiotics. In the meantime, DNA complexes were produced using 10 μg of plasmid diluted in 1 ml DMEM without serum and antibiotics. 16.6 μl of TurboFect transfection reagent were added to the diluted plasmid solution and the final

solution mixed immediately. After 15 minutes of incubation at room temperature, the DNA complexes were added to the cells. The culture medium was changed after two hours to pre-warmed DMEM complete medium.

## Cell immobilization

For correlative imaging HeLa cells were seeded in 10 cm cell culture dishes at a confluency of approximately 70%. For transfection experiments, cells were detached on the day of freezing (24 hours post-transfection) using 0.25% Trypsin in PBS and collected in a 15 ml Falcon-tube. For experiments using potential-sensitive dyes, the cells were incubated on the day of freezing with a final concentration of 200 nM MitoTracker[TM] DeepRed (ThermoFisher Scientfic, Germany) in DMEM complete medium for 1 hour. Cells were then detached by using 0.25% Trypsin in PBS and collected in a 15 ml Falcon-tube. Trypsin was deactivated by adding the double volume of pre-warmed DMEM complete medium. Cells were then centrifuged at 300 g for 5 minutes. Cell pellets were re-suspended in DMEM without phenol red and were supplemented with a final concentration of HEPES at 25 mM, before being transferred to a 1.5 ml Eppendorf-tube. The cell suspension was kept in a heated metal block at 37°C throughout this process. Prior to freezing, cells were concentrated with a table top centrifuge to achieve a paste-like consistency of the suspension. Small aliquots of about 2 μl were transferred to gold-coated copper planchettes (Engineering Office M. Wohlwend GmbH; Sennwald, Switzerland) and immobilized by high pressure freezing in a Leica HPM100 (Leica Microsystems GmbH, Wetzlar, Germany). The obtained frozen cell pellets were kept in liquid nitrogen until further use.

## Freeze substitution and resin embedding

Sample blocks were obtained by freeze substitution of the cell pellets in 0.5% uranyl acetate diluted from a 20% stock in methanol and 3% double distilled $H_2O$ in acetone in a Leica AFS1 (Leica Microsystems GmbH, Wetzlar, Germany). Temperature was set to -130°C for the first 2 hours and then raised to -90°C with a 20°C/hr gradient. Pellets were kept at -90°C for 8 hours and the temperature was then raised to -45°C in a 5°C/hr gradient. At -45°C, the pellets were washed three times with pre-cooled pure acetone over the course of 1.5 hours. For resin infiltration, the pellets were incubated with 25, 50, 75 and 100% HM20 (Science Services, Munich, Germany); dilutions in pure acetone for 2 hours each. Pellets were kept in fresh 100% HM20 over night, while the temperature was raised to -25°C with a 5°C/hr gradient. On the next day, over the course of 8 hours, the resin was replaced three times with fresh 100% HM20 resin. Pellets were then transferred to gelatin capsules and UV polymerized over the first 48 hours at -25°C, followed by a temperature rise to 0°C with a 5°C/hr gradient. After trimming of the gelatin capsules, samples could be used immediately. However, we kept sample blocks at room temperature in the dark in a fume hood to complete polymerization. Final sample blocks can easily be checked for fluorescence preservation with a 10x air objective on an upright fluorescence microscope. Alternatively, thick sections of 400 nm can be cut with a Histo-knife and observed with a fluorescence microscope.

## Fluorescence microscopy

Fluorescence images for correlative microscopy were recorded on a Leica DM6000B microscope (Leica Microsystems GmbH, Wetzlar, Germany), equipped with a CCD-camera (DFC350FX) and the following filter cubes.A4 (UV): Exc 360/40, Dichro 400, Suppres BP 470/40. GFP (blue): Exc 470/40, Dichro 500, Suppres BP 525/50. SFRED (red): Exc HQ 630/20x, Dichro Q649LP, Suppres HQ 667/30. Furthermore, this microscope allows imaging in brightfield and phase contrast microscopy.

## Electron microscopy

Electron micrographs were acquired on a Philips CM120 transmission electron microscope operated at 120 kV and equipped with a LaB$_6$-source and a TVIPS 2x2 slow-scan CCD camera.

## High-accuracy CLEM

For high-accuracy CLEM, sections of 160 nm thickness were cut with a 35˚ DiATOME ultra knife and collected on carbon-coated formvar finder grids (Ted Pella 01910-F; Electron Microscopy Sciences, Hatfield, PA, USA). Grids were kept in the dark throughout the whole process and were observed as soon as possible. For fluorescence microscopy, the grids were placed on a glass microscope slide, covered with a drop of PBS and covered with a glass cover-slip. Areas of interest were identified with a 20x air objective and more detailed images acquired with a 100x oil immersion objective. Bright field images were acquired to allow for a re-identification of the areas, DAPI channel images were acquired to image uranyl acetate fluorescence, for an easier re-identification of the cells. Finally, the channel of interest was imaged, to reveal the protein/label. After imaging, the grids were recovered by pipetting fresh PBS on one side of the cover slips, in order to float them off gently. Grids were then washed multiple times on drops of double distilled water, dried and stored until TEM investigations. Overview images were taken at an original magnification of 600x, for identification of the cells of interest. Detailed images of grid boxes were taken in a tile scan of 10x10 images, with an original magnification of 3500x. Overview images of individual cells were obtained by merging the corresponding images with Photoshop CS6. CLEM overviews were obtained by correlating the light microscopy data with the electron microscopy images in Icy with the plug-in eC-CLEM [28]. A detailed procedure and introduction for landmark based correlation can be found in the documentation of the plug-in.

## NanoSIMS imaging

To load the grids on the nanoSIMS, a sub-holder with space for three TEM grids of 10mm (#45639345, Cameca, Gennevilliers, France) was used. NanoSIMS imaging was then performed using a nanoSIMS 50L (CAMECA, Gennevilliers, France) with an 8kV $^{133}$Cs$^+$ primary ion source. The detectors were set to collect the following secondary ions: $^{12}$C$_2^-$, $^{12}$C$^{13}$C$^-$, $^{12}$C$^{14}$N$^-$, $^{12}$C$^{15}$N$^-$, $^{31}$P$^-$ and $^{32}$S$^-$. The mass resolving power was adjusted to ensure the discrimination between the peaks $^{13}$C$^{14}$N$^-$ and $^{12}$C$^{15}$N$^-$, and between $^{12}$C$^{13}$C$^-$ and $^{12}$C$_2$H$^-$. To reach the steady state of the secondary ion yield, the areas of interest were first implanted with a primary ion current of 15pA for 1 minute (primary aperture D1:1), and subsequently a primary ion current of 1.5 or 0.5pA was applied during the imaging (primary aperture D1:3 and D1:4 respectively), with an accumulation time of 5.07ms/pixel. Two consecutive layers of 512x512 pixels were obtained, with a raster size from 10x10μm to 28x28μm. Image exportation and drift correction, were performed by the OpenMIMS plugin from Fiji (http://nano.bwh.harvard.edu).

## Image registration

Precise correlation of the data obtained in the three imaging modalities can be achieved since all imaging techniques include characteristic features and intrinsic landmarks of the cells, without the need for additional fiducial markers (e.g. fluorescent beads). Image registration was carried out by monitoring a target point while picking land marks, evenly spread out throughout the field of view, until a minimal localization error was achieved for that point (2D

linear transformation). For high-accuracy CLEM the fluorescent data is up-scaled to match the electron micrograph. For detailed ion abundance, the electron micrographs as well as the fluorescent data were scaled accordingly to match the nanoSIMS data.

## Supporting information

**S1 Fig. An analysis of the same objects in different imaging modalities.** a) The same cell was imaged, from left to right, in nanoSIMS, TEM, and LM. Scale bar: 5 μm. b) Line profiles were drawn across the mitochondrion indicated by the color lines in panel a, and were fitted to Gaussian curves, to determine the full width at half maximum (FWHM) of the objects. c) An analysis of FWHM in mitochondria selected in multiple cells. For simplicity, the FWHM was left in number of pixels. No statistically significant differences could be detected using a Kruskal-Wallis test (p = 0.3384). The middle line in the boxes indicates the median, while the boxes show the 25th percentiles; the error bars show the full range of values percentiles, with symbols indicating all measurement values.
(TIF)

**S2 Fig. Lateral resolution of the CLEM-SIMS technique.** a) An analysis of the fluorescence imaging resolution. Line profiles were drawn across small spots in the MitoTracker images (same image as in Fig 4). The full width at half maximum (FWHM) was then determined from Gaussian curves fitted on the spots. The average resolution is 321 ± 36 nm (mean ± SD, from 10 different measurements). Scale bar: 5 μm. b) A similar analysis for TEM images. To determine the lateral resolution, the distance in which the intensity of the signal drops from 84% to 16% of the maximum is calculated. The average resolution is 22 ± 6 nm (mean ± SD, from 10 different measurements). Scale bar: 500 nm. c) A similar analysis for SIMS images, relying on $^{12}C^{14}N$ images, and using the 84%-16% resolution measurement (same image as in Fig 4). The average resolution is 164 ± 27 nm (mean ± SD, from 7 different measurements). Scale bar: 5 μm.
(TIF)

**S3 Fig.**
(TIF)

## Acknowledgments

We thank Katharina Grewe for help with nanoSIMS imaging.

## Author Contributions

**Conceptualization:** Felix Lange, Paola Agüi-Gonzalez, Dietmar Riedel, Nhu T. N. Phan, Stefan Jakobs, Silvio O. Rizzoli.

**Data curation:** Paola Agüi-Gonzalez, Dietmar Riedel, Stefan Jakobs.

**Formal analysis:** Felix Lange, Paola Agüi-Gonzalez, Silvio O. Rizzoli.

**Funding acquisition:** Stefan Jakobs, Silvio O. Rizzoli.

**Investigation:** Felix Lange, Paola Agüi-Gonzalez.

**Methodology:** Felix Lange, Paola Agüi-Gonzalez, Dietmar Riedel, Nhu T. N. Phan, Stefan Jakobs, Silvio O. Rizzoli.

**Project administration:** Stefan Jakobs, Silvio O. Rizzoli.

**Resources:** Stefan Jakobs, Silvio O. Rizzoli.

**Software:** Paola Agüi-Gonzalez, Silvio O. Rizzoli.

**Supervision:** Stefan Jakobs, Silvio O. Rizzoli.

**Validation:** Felix Lange.

**Writing – original draft:** Felix Lange, Paola Agüi-Gonzalez, Stefan Jakobs, Silvio O. Rizzoli.

**Writing – review & editing:** Felix Lange, Paola Agüi-Gonzalez, Dietmar Riedel, Nhu T. N. Phan, Stefan Jakobs, Silvio O. Rizzoli.

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
