## [Decision Letter · Decision Letter 0]

13 Nov 2020

PONE-D-20-30666

Correlative fluorescence microscopy, transmission electron microscopy and secondary ion mass spectrometry (CLEM-SIMS) for cellular imaging

PLOS ONE

Dear Silvio,

Thank you for submitting your manuscript to PLOS ONE. After careful consideration, we feel that it has merit but does not fully meet PLOS ONE’s publication criteria as it currently stands. Therefore, we invite you to submit a revised version of the manuscript that addresses the minor points raised by one of the reviewer. Espacially, the methods section needs to be better developed as it lacks some information.

We look forward to receiving your revised manuscript.

Kind regards,

Stephane Gasman

Academic Editor

PLOS ONE

Journal Requirements:

Reviewers' comments:

Reviewer's Responses to Questions

**Comments to the Author**

1. Is the manuscript technically sound, and do the data support the conclusions?

Reviewer #1: Yes

Reviewer #2: Yes

2. Has the statistical analysis been performed appropriately and rigorously? 

Reviewer #1: Yes

Reviewer #2: I Don't Know

3. Have the authors made all data underlying the findings in their manuscript fully available?

Reviewer #1: Yes

Reviewer #2: Yes

4. Is the manuscript presented in an intelligible fashion and written in standard English?

Reviewer #1: Yes

Reviewer #2: Yes

5. Review Comments to the Author

Reviewer #1: The Authors are presenting a protocol that enables the correlation between three imaging techniques. Starting from a fully established CLEM protocol based on sequential imaging with Light Microscopy (LM) and Electron Microscopy (EM) of thin resin sections of vitrified/freeze substituted resin embedded material, the Authors succeeded in analyzing the same sections with secondary ion mass spectroscopy (SIMS). The resulting method, CLEM-SIMS, offer the possibility to overlay to the EM ultrastructure the compositional information gained by nanoSIMS having also the chance to target specific cells or organelles due to the correlation with LM.

As proof of principle the Authors imaged in a correlative manner cultured cells stained with MitoTracker or mito-mCitrine transfected. CLEM-SIMS approach showed that non-functional mitochondria (i.e. MitoTracker negative) despite having a similar morphology, have a molecular composition different from functional ones. This clearly shows the advantage of correlating analytical information with a CLEM approach over an independent correlation between SIMS and LM or EM alone.

I agree with the authors that their protocol will be extremely useful to correlating SIMS signal to specific cellular structure and potentially to study turnover of proteins and molecule.

I have only few technical concerns.

1)Uranyl acetate fluorescence

The Authors claimed that the fluorescent signal observed in the DAPI channel is the signal from uranyl acetate ( see lines 122-123; 130 and 435). In a recent publication Tuijtel, M. W. et al. (DOI: 10.1038/s41598-017-10905-x) used indeed uranyl acetate (UA) fluorescence in order to correlate CLEM images. Instead of what the Authors are claiming Tuijtel, M. W. et al. reported that the fluorescent signal coming from UA is visible using filter set for GFP (excitation 470/40, emission 525/50) at cryogenic temperature, and far less intense at 21°C. Moreover the fluorescent signal arising from UA revealed the outline of the cells, nucleoli and mitochondria. In figure 2 of this manuscript instead the DAPI channel is highlighting heterochromatic regions. Could the authors comment on these discrepancies?

2)Material and Methods

The authors should reported the excitation and emission parameter of the fluorescent filter sets and the imaging parameter used. Information of the type of transmission electron microscope used, camera and pixel sizes are missing.

Image registration

In the image registration chapter, the author should explain in more detail the procedure of alignment and ideally give an estimation of the alignment accuracy over the 3 imaging modalities. As suggestion the Authors could draw a line on structures visible in the 3 modalities (i.e mitochondria) calculate the full-with-half maximum of the intensity over the line for EM, LM and nanoSIMS in order to obtain an estimation of the alignment accuracy. Such images with their respective plots calculated from the signal of the 3 images modalities could be added as supplementary data.

Minor comments.

High pressure freezing/freeze substitution (HPF/FS).

The authors correctly decide to avoid chemical fixation and process their sample with HPF/FS as they explained in the Discussion chapter. Nevertheless, in order to process the samples in HPF planchettes, the cells were detached using trypsin. In my opinion for further studies the Author should consider to use sapphire disc in order to vitrify cultured cells without the need of tripisinization.

NanoSIMS compare to other analytical methods.

Other analytical methods have been recently developed in the contest of a correlative approach. The author did a good effort in highlighting potentiality and pitfall of the technology they developed. For a reader would be useful also to compare this new methods with other correlative approached based on other analytical techniques (i.e. energy dispersive X-ray (EDX) or electron energy loss spectroscopy (EELS)).

Reviewer #2: This is a very interesting and useful manuscript in which the authors present a novel and insightful methodology to correlate the ultrastructure of cells with the molecular composition of cell compartments. The technique is based in previous methodological advances achieved by the group of Dr. Silvio Rizzoli who leads this study. The novelty is the combination of electron microscopy and fluorescence microscopy with secondary ion mass spectrometry (SIMS) to visualize the localization of spots enriched in different types of natural isotopes, which is useful to, for example, detect specific organelles (such as mitochnodria o difference in chromatin organization). The group is pionneering here the implementation of high pressure freezing which is an advantage to investigate cellular ultrastructure under conditions that are better to preserve structures compared to conventional fixation methods. This is a methodological and useful study in which the conclusions are supported by high quality data. The authors explain and discuss the methods in detail. The paper is very nicely and clearly written and opens new perspectives to investigate specific aspects of the biology of the cell such as the aging of organelles.

6. PLOS authors have the option to publish the peer review history of their article (what does this mean?). If published, this will include your full peer review and any attached files.

Reviewer #1: No

Reviewer #2: No

---

## [Author Response · Author response to Decision Letter 0]

9 Mar 2021

Reply to Reviewers

Reviewer #1: 

The Authors are presenting a protocol that enables the correlation between three imaging techniques. Starting from a fully established CLEM protocol based on sequential imaging with Light Microscopy (LM) and Electron Microscopy (EM) of thin resin sections of vitrified/freeze substituted resin embedded material, the Authors succeeded in analyzing the same sections with secondary ion mass spectroscopy (SIMS). The resulting method, CLEM-SIMS, offer the possibility to overlay to the EM ultrastructure the compositional information gained by nanoSIMS having also the chance to target specific cells or organelles due to the correlation with LM. 

As proof of principle the Authors imaged in a correlative manner cultured cells stained with MitoTracker or mito-mCitrine transfected. CLEM-SIMS approach showed that non-functional mitochondria (i.e. MitoTracker negative) despite having a similar morphology, have a molecular composition different from functional ones. This clearly shows the advantage of correlating analytical information with a CLEM approach over an independent correlation between SIMS and LM or EM alone.

I agree with the authors that their protocol will be extremely useful to correlating SIMS signal to specific cellular structure and potentially to study turnover of proteins and molecule.

I have only few technical concerns.

We thank the Reviewer for the comments.

1)Uranyl acetate fluorescence

The Authors claimed that the fluorescent signal observed in the DAPI channel is the signal from uranyl acetate ( see lines 122-123; 130 and 435). In a recent publication Tuijtel, M. W. et al. (DOI: 10.1038/s41598-017-10905-x) used indeed uranyl acetate (UA) fluorescence in order to correlate CLEM images. Instead of what the Authors are claiming Tuijtel, M. W. et al. reported that the fluorescent signal coming from UA is visible using filter set for GFP (excitation 470/40, emission 525/50) at cryogenic temperature, and far less intense at 21°C. Moreover the fluorescent signal arising from UA revealed the outline of the cells, nucleoli and mitochondria. In figure 2 of this manuscript instead the DAPI channel is highlighting heterochromatic regions. Could the authors comment on these discrepancies?

As the Reviewer points out, the work from Tuijtel and collaborators employed uranyl acetate fluorescence at cryogenic temperatures. This procedure not only enhances the fluorescence of uranyl acetate, but also changes its emission spectrum (Jones et al., Chem Sci, 2015, 6:513). At room temperature a peak appears in the spectrum covered by the DAPI channel, which is measurable using a conventional microscope. This is more easily measured in our work than in the work of Tuijtel and colleagues, since our sections are substantially thicker.

The fluorescence of the uranyl acetate in the green channel (the one employed by Tuijtel and collaborator) should still be measurable at room temperature, according to Jones et al., Chem Sci, 2015, 6:513. A small signal can be detected, as suggested by Tuijtel and collaborators, but it is so low as to be negligible in comparison to specific GFP signals (please see Figure 1 of our manuscript).

The localization of the uranyl acetate fluorescence is not different, in our opinion, between the two studies. We also find nucleoli to be well-labeled, in all cells, and the heterochromatin seems to be well labeled in the work of Tuijtel and collaborators (please see their Figure 3). Mitochondria also are labeled in our work, albeit the signal is dim. As an example, please find below a view in which we do not show the MitoTracker or GFP channel, and we use the same colormap as Tuijtel and colleagues. The resulting image is very similar to the published work of Tuijtel and colleagues.

(Find figure enclosed in "Response to Reviewers" file) 

Figure for reviewers. Uranyl Acetate in DAPI channel. a) Uranyl acetate fluorescence in 160 nm thick resin sections of HeLa cells imaged in the DAPI channel, using a green colormap. b) The same image of the DAPI channel displayed in grayscale with inverted look up table (LUT). c) Transmission electron micrograph of the same region on the resin section imaged at 120 kV with 3500x original magnification. Scale bars: 5µm.

2)Material and Methods

The authors should reported the excitation and emission parameter of the fluorescent filter sets and the imaging parameter used. Information of the type of transmission electron microscope used, camera and pixel sizes are missing.

We have now added these missing parameters.

Image registration

In the image registration chapter, the author should explain in more detail the procedure of alignment and ideally give an estimation of the alignment accuracy over the 3 imaging modalities. As suggestion the Authors could draw a line on structures visible in the 3 modalities (i.e mitochondria) calculate the full-with-half maximum of the intensity over the line for EM, LM and nanoSIMS in order to obtain an estimation of the alignment accuracy. Such images with their respective plots calculated from the signal of the 3 images modalities could be added as supplementary data.

We have now performed this type of analysis, by measuring the full-with-half-maximum (FWHM) for several mitochondria that are labeled in all channels. The resulting values are similar, in spite of differences in the resolutions of the three techniques. We have now added the respective figure as Supplementary Data.

Minor comments.

High pressure freezing/freeze substitution (HPF/FS).

The authors correctly decide to avoid chemical fixation and process their sample with HPF/FS as they explained in the Discussion chapter. Nevertheless, in order to process the samples in HPF planchettes, the cells were detached using trypsin. In my opinion for further studies the Author should consider to use sapphire disc in order to vitrify cultured cells without the need of tripisinization.

We will use this technique in the future. We have now made a comment on this approach in our Discussion.

NanoSIMS compare to other analytical methods.

Other analytical methods have been recently developed in the contest of a correlative approach. The author did a good effort in highlighting potentiality and pitfall of the technology they developed. For a reader would be useful also to compare this new methods with other correlative approached based on other analytical techniques (i.e. energy dispersive X-ray (EDX) or electron energy loss spectroscopy (EELS)).

We have added a paragraph on these important techniques in the Discussion section.

Reviewer #2:

This is a very interesting and useful manuscript in which the authors present a novel and insightful methodology to correlate the ultrastructure of cells with the molecular composition of cell compartments. The technique is based in previous methodological advances achieved by the group of Dr. Silvio Rizzoli who leads this study. The novelty is the combination of electron microscopy and fluorescence microscopy with secondary ion mass spectrometry (SIMS) to visualize the localization of spots enriched in different types of natural isotopes, which is useful to, for example, detect specific organelles (such as mitochnodria o difference in chromatin organization). The group is pionneering here the implementation of high pressure freezing which is an advantage to investigate cellular ultrastructure under conditions that are better to preserve structures compared to conventional fixation methods. This is a methodological and useful study in which the conclusions are supported by high quality data. The authors explain and discuss the methods in detail. The paper is very nicely and clearly written and opens new perspectives to investigate specific aspects of the biology of the cell such as the aging of organelles.

We thank the Reviewer for the comments.

---

## [Decision Letter · Decision Letter 1]

13 Apr 2021

Correlative fluorescence microscopy, transmission electron microscopy and secondary ion mass spectrometry (CLEM-SIMS) for cellular imaging

PONE-D-20-30666R1

Dear Silvio,

We’re pleased to inform you that your manuscript has been judged scientifically suitable for publication and will be formally accepted for publication once it meets all outstanding technical requirements.

Kind regards,

Stephane Gasman

Academic Editor

PLOS ONE

Additional Editor Comments (optional):

Reviewers' comments:

Reviewer's Responses to Questions

**Comments to the Author**

1. If the authors have adequately addressed your comments raised in a previous round of review and you feel that this manuscript is now acceptable for publication, you may indicate that here to bypass the “Comments to the Author” section, enter your conflict of interest statement in the “Confidential to Editor” section, and submit your "Accept" recommendation.

Reviewer #1: All comments have been addressed

Reviewer #2: All comments have been addressed

2. Is the manuscript technically sound, and do the data support the conclusions?

Reviewer #1: Yes

Reviewer #2: Yes

3. Has the statistical analysis been performed appropriately and rigorously? 

Reviewer #1: Yes

Reviewer #2: I Don't Know

4. Have the authors made all data underlying the findings in their manuscript fully available?

Reviewer #1: Yes

Reviewer #2: Yes

5. Is the manuscript presented in an intelligible fashion and written in standard English?

Reviewer #1: Yes

Reviewer #2: Yes

6. Review Comments to the Author

Reviewer #1: (No Response)

Reviewer #2: (No Response)

7. PLOS authors have the option to publish the peer review history of their article (what does this mean?). If published, this will include your full peer review and any attached files.

Reviewer #1: No

Reviewer #2: No

---

## [Editor Report · Acceptance letter]

30 Apr 2021

PONE-D-20-30666R1 

Correlative fluorescence microscopy, transmission electron microscopy and secondary ion mass spectrometry (CLEM-SIMS) for cellular imaging 

Dear Dr. Rizzoli:

I'm pleased to inform you that your manuscript has been deemed suitable for publication in PLOS ONE. Congratulations! Your manuscript is now with our production department. 

Kind regards, 

on behalf of

Dr. Stephane Gasman 

Academic Editor

PLOS ONE